# Innovation in Road Freight Transport: Quantifying the Environmental Performance of Operational Cost-Reducing Practices

**Valentin Carlan \*, Christa Sys**  **and Thierry Vanelslander**

Department of Transport and Regional Economics, University of Antwerp, Prinsstraat 13,
2000 Antwerp, Belgium; christa.sys@uantwerpen.be (C.S.); thierry.vanelslander@uantwerpen.be (T.V.)

**\*** Correspondence: valentin.carlan@uantwerpen.be; Tel.: +32-3-265-49-14

**Abstract:** Road transportation is a key mode of transport when it comes to ensuring the hinterland connection of most European ports. Constrained by low profit margins and having to be active in a highly competitive market, companies active in this sector seek multi-dimensional innovative solutions that lower their operational costs. These innovative initiatives also yield positive environmental effects. The latter however are poorly recognized. This paper investigates the characteristics of different types of chassis used to transport containers from and to the terminals in the port areas and looks into the details of operational planning practices. It analyses the cost-effectiveness of these innovative solutions highlighting both the costs and the environmental emissions they save. Transport data from a road hauler serving the hinterland connection of a port in Western Europe is used to build up a case study. Results show that by using special types of chassis, which enable the combination of transport tasks in round-trips, the operational costs are reduced by 25% to 35%, and equally the $CO_2$ emissions are also decreased by 34% to 38%.

**Keywords:** road transport innovation; cost effectiveness analysis; environmental performance

## 1. Introduction

Road haulage is a key mode of transport mode when it comes to ensuring the hinterland connection of most European sea-ports. Loads such as containers, break bulk, dry bulk, or liquid bulk cargo depend on road transport as a part of their supply chain. Figure 1 compares ports' modal split in Europe and shows the dominance of road transport. As a consequence, it is clear that port traffic growth is challenging more and more the capacity of road networks developed around ports [1]. In this context, road transport companies are forced to seek their position in a highly competitive market. Equally, road freight transport is a sector with low profit margins. For these reasons, road haulers are now challenged to innovate and to find solutions to achieve cost-effective road transport solutions. In most cases, these solutions are triggered by policies on sustainable development (e.g., road pricing). However, the conceptual developed and implementation is done using private financial capabilities.

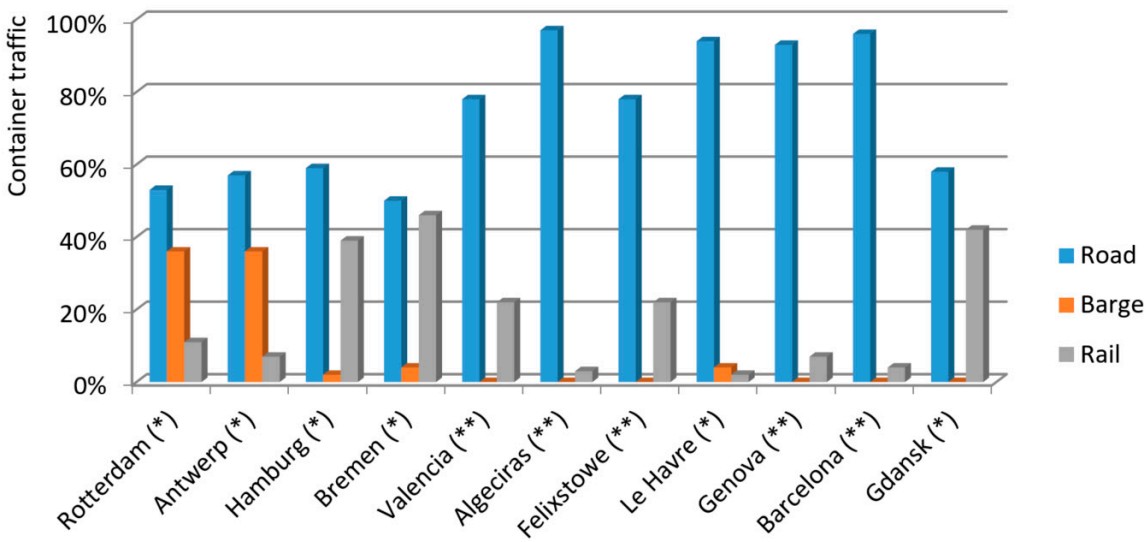

(*) 2015 traffic | (**) 2013 traffic

**Figure 1.** Modal split of port hinterland traffic in European ports. Source: [2–7].

Releases of the European Commission, such as its White paper [8], acknowledge the importance of road transport and create a clear context for its integration among all other transport modes. Moreover, European legislation is making efforts to create new ways of stimulating the use of environment-friendly solutions for transport.

In this context, road transport innovators are confronted with the task of keeping a positive balance between their operational costs and policy decisions regarding environmental emissions. In practice, even if the environmental advantages are proven, the best economic solution still rules the decisions to invest in innovation dedicated to enhance the ports' activity [9].

Ex-post evaluation studies that use real data to show the effects of innovation in this field are scarce. Giuliano et al. [10] show that only a limited set of studies exist that use methods that quantify the costs and benefits in analyses of innovation in the transport sector. Conducting analyses like cost-benefit, total cost, or cost-effectiveness requires a considerable amount of data. Information regarding investment by companies in innovation and their outcome are not always available. Reasons for this gap are the confidentiality status of this type of data. Innovation is driven by experimental thoughts and the nature of decisions in the supply chain that are sometimes made based on gut feeling [9], for which recordings are private. The latter brings limitation in methods and studies that can be developed.

Nakamura [11] notes that the roots of scientific evaluation of projects in the transport sector dates back to the 19th century. Since then, different types of methods and forms of evaluation have been developed to shed light on different aspects of effectiveness for investment and innovation projects developed for the transport sector. For this matter, methods following descriptive, qualitative or quantitative approaches have been developed and applied for the analysis of innovation in the transport sector.

The present research develops an application of the cost-effectiveness analysis (CEA) to investigate whether the innovative practices used in road transport are bringing both operational and environmental cost advantages. The CEA is adapted to analyze decisions in which expected outcomes are clearly identified, but whose direct or indirect monetary benefits are not easily quantifiable [12]. CEA has a long tradition that dates back in 1960s, when scientists developed a method to assist the United States military in making allocation decisions [13]. This method, among others, has later been successfully applied in the medical sector. Here, the benefits of decisions or interventions could not be quantified more than the number of lives saved or persons cured [14–16].

Staying competitive is the main objective of road transporters. In order to do so, developing cost-effective innovation is seen as the future of transport hauliers. Academia [17–19] provides multiple definitions of innovation. The present research follows the definition provided by Arduino et al. [20], who identify innovation as "the technological or organizational (including cultural and marketing as separate subsets) change to the product (or service) or production process that either lowers the cost of the product (or service) or production process or increases the quality of the product (or service) to the consumer". From a logistics chain and port-related perspective, Vanelslander et al. [21] show that stakeholders encompass multi-dimensional innovation and puts forward a comprehensive list of innovation types. Their typology (shown in Appendix A) is applicable as well to innovation developed by road transport operators.

Blauwens et al. [22] put forward three classical examples of reducing the operational cost in a road transport firm. The first one refers to determining the shortest routes for its vehicles. Another practice is performing round trips that combine two or more transport requests in one journey. The third way of reducing transport cost is assigning transport tasks to vehicles that have the starting point as close as possible to the end location of a previous one. These three solutions set the goal to minimize the total distance that the transport vehicles are traveling per day. One should bear in mind that the total "distance", depending on the scope of the optimization problem, can be also translated into time or cost, besides kilometers [22]. To set it in practice, road transporters together with chassis constructors look for innovative solutions. This cooperation resulted in the development of new trailers (or chassis) that extends the range of container transport tasks that can be consolidated in one trip. As a result, road hauliers' cost effectiveness is improved. The amount of kilometers driven in productive trips are increased and the labor time gets decreased. Simultaneously, the use of these newly-developed chassis led to perform transport tasks that also reduce environmental emissions. These results are enabled through both an innovative planning process and innovative chassis development. These changes are introduced through a new managerial and cultural (change of mind set) approach. In essence, this is thus an example of multi-dimensional innovation that falls under innovation type III (technological changes that also consist of changes at managerial, organizational and cultural level for a specific business) as put forward by Vanelslander et al. [21]. Although these solutions are in use and known to the stakeholders active in this sector, this innovation in road transport does not get immediately adopted by the entire road transport sector and it remains unknown to the wide public. For this reason, the road transport stakeholders often get the label of un-innovative sector.

The results of this multi-dimensional innovation are analyzed by this study through a CEA. This research quantifies the amount of kilometers that vehicles drive, the working hours for the employees that are active, the tolls that are paid or the vehicle usage cost inquired. These elements are finally used as costs that a transport haulier saves when innovative chassis are in use. The latter emissions savings are considered as extra benefits resulting from the more cost-effective activity.

The empirical analysis developed by this study reflects the perspective of a road haulier active in container transport. In this setting, the focus is on transport of 20' import/export containers within the hinterland area of a seaport, as this specific setting enable container transport task consolidation in round trips.

The research questions addressed by this study are twofold:

RQ1: What is the cost-effectiveness of the innovative chassis-fleet management practice adopted by a road transport firm in order to minimize its operational costs? and
RQ2: What is the cost effectiveness of these solutions with respect to environmental emissions?

To answers these questions, the following paper structure is pursued. Section 2 presents the technical innovation present in road transport and the cost-effectiveness analysis as the main method used to achieve the results of these study. The latter are detailed in Section 3. Finally, Section 4 concludes this chapter.

## 2. Material and Methods

This section provides a detailed description of the multi-dimensional innovative solutions analyzed by this research and it delivers the background of the CEA.

### 2.1. Innovative Practices in Road Transport of Containers

This research focuses thus on pointing out the effect of efficient use of innovative trailers. To do so, the practice of carrying out individual transport tasks is firstly explained. The subsequent sub-subsections detail the technical characteristics of innovative chassis, the conditions considered when carrying containers via road and the decision algorithm of setting up round-trips.

#### 2.1.1. Carrying out Individual Transport Tasks

The working practice considered by this analysis as a reference scenario in carrying out transport tasks has the following characteristics. Firstly, the transport company does not have the tools to comprehensively bundle data with regard to its transport orders in a centralized data-base. This type of working practice is typical for a company that uses multiple communication channels (e.g., email requests, phone, and shipping agents' platforms) to collect orders and their related data. Secondly, it is assumed that a transport company does not have the technical capacities to carry out more than one transport task at once. This characteristic applies to road transport of 20' containers where one trailer can accommodate only one container.

Considering the above characteristics, a transport company conducts the following working practice to deliver its orders. This working practice is referred to as the reference scenario by the current research. For each transport order, shown in Figure 2 with a full line, the transport company completes two trips: one trip corresponding to the transport order and another trip corresponding to the empty return of the transport equipment (trailers). This situation occurs for both orders either to be carried from or towards point *A*. The extra time and the extra fuel consumption that is used to relocate the truck to its origin needs to be reduced. This reference scenario is being used as a comparison situation for the other alternative solutions chosen to carry the full transport tasks.

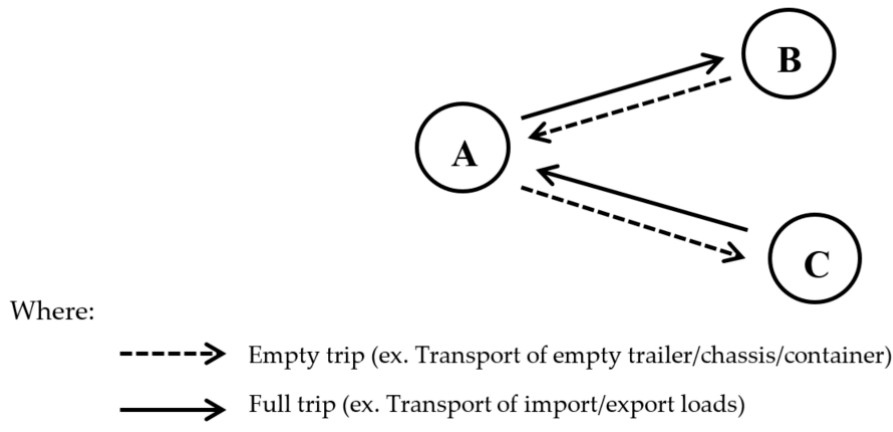

**Figure 2.** Carrying individual transport tasks.

#### 2.1.2. Multi-Dimensional Innovation Set in Use by Road Haulers for Round-Trips

Transport tasks are characterized by the following elements: origin/destination, time of pick-up/drop-off, type of container and weight. These are the basic details needed by a transport planner to estimate which tasks can be carried out in a combined round-journey and which type of chassis should be used for this combination of tasks. The type of assets and the type of round-journeys enabled through multi-dimensional innovation are detailed below. These options are applicable when 20' containers or swap bodies similar in size without side doors are transported and when any

transhipment or container swap movement with equipment (e.g., cranes, reach stackers) at hinterland locations (points B or C) are unavailable.

### A.    Using Mini Eco-Combi Chassis

Under 'mini eco-combi cassis' is understood a combination of two 20' chassis that can be attached together so as to form a single 40' unit (Figure 3). The advantage of this type of chassis is that, depending on the round-journey characteristics, the two chassis can be also used as individual transport units. Equally, the mini eco-combi with its flexibility allows transporters to consider more combinations of journeys in their planning operations, eliminating technical limitations (conditions). The conditions imposed by safety regulations with regards to weight distribution for road transport vehicles are set by the European Commission (2015). According to EC 2015/719, the traction axle of the towing tractor must carry a minimum of 25% of the total carried load. The decoupled chassis enable the transporters to carry further to its destination a fully loaded 20' container that was previously loaded on the rear side of the chassis. Involving a regular 40' chassis in this practice, would defy the regulation. A further explanation of containers weight distribution is later given in Table 1. A derived advantage of this type of chassis is the time saved due to faster actions done to couple/uncouple the two chassis.

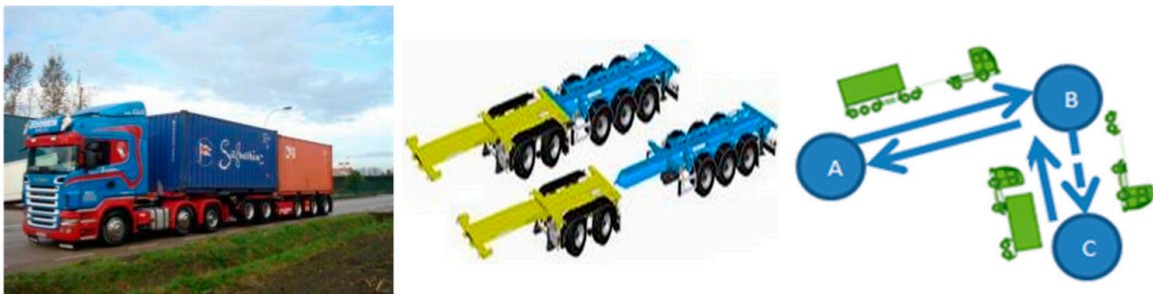

**Figure 3.** Mini eco-combi chassis. source: [24,25].

### B.    Operating Multipurpose Chassis

The multipurpose chassis is used to complete combinations of two transport tasks in one round-trip. The multipurpose chassis category refers to round transport journeys carried out with the following type of chassis: steer-chassis, legs-mounted chassis, and hydraulic load-shifting chassis. The use of these solutions is not as flexible as the mini eco-combi chassis due to a fixed weight distribution and loading/unloading order constraints. Nevertheless, these options are still suitable for combining different transport tasks in one trip, as shown in Figure 4. In addition, the amount of time saved by using this alternative is lower than in the case of mini eco-combi chassis, due to extra operations that have to be done at each loading/unloading point. Larger time windows used to position the containers according to safety procedures are counted in. For example: the new steer-chassis needs extra time to reposition the container from the back end of the chassis to the front and to extend the steering third axle; the leg-mounted chassis operations need extra time to expand/retract the mounted legs, to set in use the pneumatic suspension of the chassis and reposition the container on the chassis; the classical chassis requires extra equipment (e.g., crane or reach stacker) to be used for the container to be repositioned.

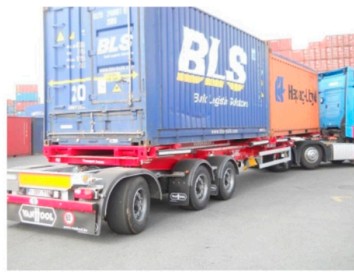
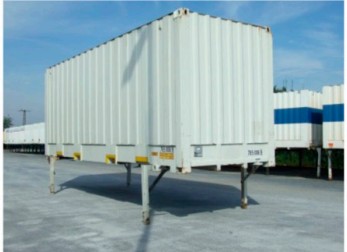
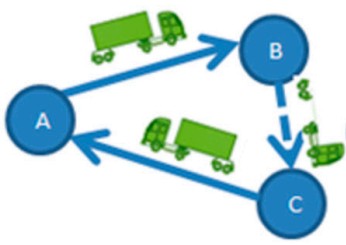

**Figure 4.** Multipurpose chassis use. Source: [25].

### C. Offering Empty Containers for Re-Use

The 're-use' of empty containers has as main advantage the extra financial gain that the company makes from offering an empty container to be re-filled. This practice is driven through the innovative mind set on managing a business unit with multiple technological solutions. Although the re-use of empty containers seems to be an extra income, a certain balance has to be achieved between the benefit of the re-use and the extra fees charged by container owners, which commit to this option. The route created in a re-use round-journey is similar to the one presented in Figure 4.

### D. Handling Transport Tasks with Tilt Chassis

The tilt chassis (Figure 5) represents another alternative that joins the innovative practices, which reduces the costs for pick-up/dropping cargo in road transport. This type of chassis enables the planner to form round-trips that reduce the distance travelled. A tilt chassis is a fast solution for cargo to be unloaded. Not all containers can be unloaded using a tilt method; however, this type of chassis increases the chances of creating a combined round-trip by offering the same container to be re-filled.

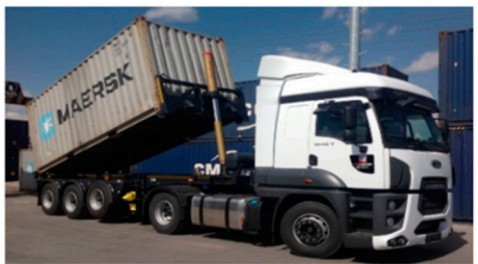
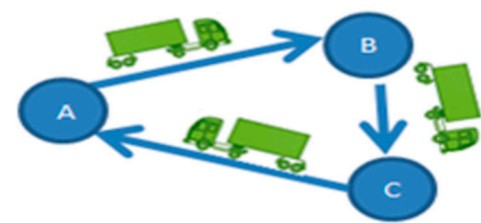

**Figure 5.** Tilt chassis. Source: [25].

### 2.1.3. Operational Conditions Considered when Combining Single Transport Tasks

The weight of each load is a key criterion in the process of creating round journeys. Due to road tonnage restrictions and road safety regulation, several conditions must be taken into account. Table 1 shows the possibilities for two 20' containers to be loaded on chassis in accordance with the directives given by the European Commission (2015). The transport planner must take these conditions into account when the transport tasks are attributed to each round-journey. The maximum load is calculated by adding the truck, chassis, container and cargo weight.

**Table 1.** Loading conditions considered when creating round journeys. Source: own composition based on interviews.

| | Condition | Observations |
|---|---|---|
| **Combinations of round-journeys that respect the following conditions are allowed** | | |
|  | $F_1 + F_2 <$ Max load [1] | Two loaded containers, the total weight of which is not higher than the total allowed weight |
|  | $F_1 + E_2 <$ Max load<br>$E_1 + F_2 <$ Max load<br>$E_1 + E_2 <$ Max load | One loaded container and one empty container the total weight of which is not higher than the total allowed weight. Or two empty containers bearing the same condition. |
|  | 25%*$F_1$ = Load on traction axis | One loaded container of which at least 25% of its weight is distributed on the traction axle. |
| **Combinations of round-journeys that lead to one of the following situations are not allowed** | | |
|  | $F_1 + F_2 >$ Max load | Two loaded containers, the total weight of which is higher than the total allowed weight. |
|  | 100%*$F_1$ = Load on rear axle | One loaded container the total weight of which is distributed on the rear chassis axles. |

Note: 1. Maximum allowed weights for road freight transport according to each European country's regulation and in accordance with Directive (EU) 2015/719 [23].

The use of each chassis is conditioned by the order in which the transport loads are delivered. Table 2 shows the types of round journeys than can be performed by each type of chassis. A parallel comparison between classical, multipurpose and mini eco-combi chassis has been made. Moreover, a separate column is added for the re-use of containers and the use of tilt chassis.

The following notations are being used to define the types of possible round transport journeys by using each chassis, where *P*, *D* and *Pa* are notations used for operations involving classical, multipurpose or eco-combi chassis respectively, and *L* and *U* are used for operations involving re-use of containers and tilt chassis where cargo is loaded respectively unloaded from the carried containers.

*P*—pick (or loading) of a container on a chassis;
*D*—drop (or unloading) of a container from a chassis;
*Pa*—park or decoupling of chassis and park;
*L*—loading a container;
*U*—unloading a container.

As can be seen in Table 2, the alternative solutions for carrying out transport requests are divided in three categories. These categories are created based on the characteristics of each chassis and the type of round-trips that they can perform:

- Round-transport tasks using multipurpose chassis (includes chassis which allow loading two 20-foot containers—one export and one import, a leg-mounted chassis and a new steer-chassis);
- Round-transport tasks using the mini eco-combi chassis; and
- Round-transport tasks using tilting chassis (re-using the same 20 containers).

**Table 2.** Possible transport journey combinations. Where the following notations are used: PD—pick-drop; PDPD—pick-drop-pick-drop; PULD—pick-unload-load-drop; PPDD—pick-pick-drop-drop; PPaDPD—pick-park-drop-pick-drop. Source: own composition based on interviews.

| | Type of Combined Trip | Classical Chassis | Multipurpose Chassis | | | Mini Eco-combi | Tilt Chassis and Re-use |
| --- | --- | --- | --- | --- | --- | --- | --- |
| | | | Rear Wheel Steering Chassis | Legged Mounted Chassis | Hydraulic Chassis | | |
|  | PD | ✓ | ✓ | ✓ | ✓ | ✓ | ✓ |
|  | PDPD PULD | ✓ | ✓ | ✓ | ✓ | ✓ | ✓ (PULD) |
|  | PPDD (1) | | ✓ | ✓ | ✓ | ✓ | |
|  | PPDD (2) | | ✓ | ✓ | ✓ | ✓ | |
|  | PPDD (3) | | ✓ | ✓ | ✓ | ✓ | |
|  | PPaDPD | | | | | ✓ | |

## 2.2. Cost-Effectiveness Analysis

The main purpose of a CEA is to identify the economically most efficient way of fulfilling an objective [26]. The European Commission agrees with this type of analysis for evaluations of programs or projects, to assess the choices in the allocation of resources, to determine strategy-planning priorities or to be used in argumentative debates. CEA can be conducted in the context of ex-ante, intermediary, or ex-post evaluations. Contrary to the CBA monetizing practice, CEA puts in balance two elements: the cost of achieving one objective and the level of achievement of that objective. In other words, a cost effectiveness analysis makes a ratio between the inputs in monetary terms and the outcomes in non-monetary quantified terms [27]. Input data for the CEA, as well as for the CBA, is difficult to collect. The main reason for this difficulty is the availability of this type of data and their confidentiality status. Nevertheless, applying CEA has its advantages. CEA is a tool that assesses a decision by using a single dimension of its output. CEA can measure the technical effectiveness of a project due to its particularity of comparing the costs of a project with its immediate outputs. Furthermore, CEA can also be used as a comparison tool of several projects or investment options. For this case, it must be taken into consideration that the evaluated criteria should be the same over the entire range of options [26].

### 2.2.1. CEA: Outcomes from a Literature Review of Transportation Studies

A non-exhaustive literature review was conducted, focusing on studies using CEA for transport applications in the 2000–2015 period. From a transport mode perspective, it is clear that the road

transport sector benefits from a lot of attention from researchers applying CEA. The scope of studies in this case was to prove the effectiveness of different measures either on social or environmental matters [28,29]. Furthermore, also the rail sector benefits from the attention of CEA applications. The review of studies shows that the scope of CEA in this case was mixed, being focused on either social, economic or environmental matters [30–32]. Hence, applications of CEA have also been used to determine the effect on environmental matters of maritime strategies. Here is to be mentioned the effect of speed limitations imposed on maritime transport emissions [33–35] or the consequences of policies regarding container repositioning [35]. CEA was not preferred in analyzing matters dedicated to air transport. The authors of [36] use CEA only as an extension of a wider CBA to point out the cost effectiveness of measures taken by several airports against terrorist attacks. Their findings indicate that any additional measures against terrorist attacks would be too expensive to be justified by their effectiveness.

The literature review shows that the scope of CEA in transport studies is to determine the effectiveness of measures for which a concrete estimation of all its benefits is difficult to determine [27,37]. For this reason, most CEA analyses concern environmental impact assessments [28,32,38,39] or quantification of costs for measures addressing social benefits in the short term [29]. In this case, researchers quantify the average cost of emission mitigation measures or actions aimed at decreasing the number of road fatalities. In some cases, CEA can also use monetary values as outcome indicators; these studies have a purely economic purpose [30,40,41].

The CEA outcome is most of the time expressed in units of monetary value spent to achieve one unit of abatement measure. Depending on the purpose of each analysis, the units used as measurements of effectiveness are chosen. More specifically, the effectiveness of pollution control measures is expressed in the average cost of the amount of emissions avoided ($CO_2$, $NO_x$, $PM_x$, or $SO_x$) [28,32–34]. In the case of measures with a social impact, the immediate outcomes are quantified in the amount of averted deaths or the value of life [29]. Some of the research studies on determining the cost effectiveness of projects in the transport sector do not end by calculating a CEA ratio. These studies are structured in three parts. Firstly, the costs of the analyzed measures are inventoried and quantified. Secondly, the implications and the immediate benefits of these measures are also calculated. Following these calculations, a discussion is being conducted based on the cost and the project outcomes previously determined [41–43].

### 2.2.2. Cost-Effectiveness Analysis Steps

This section summarizes the general steps that have to be undertaken to conduct a CEA. Figure 6 offers a brief overview of the steps that are undertaken and its specific outputs for a road transport case analysis.

In order to apply the cost effectiveness analysis, a first step is to define the framework and the scope of the evaluation. Also at this stage, the reference scenario and the alternative(s) are being described. As a second step, the costs for the reference scenario and for each alternative operational solution is calculated. In particular, for a road transport case analysis, this step refers to the calculation of operational expenses such as fuel consumption, tolls, labor cost, or other fees that are being supported by the transport company [22]. Thirdly, the benefits in terms of distance, hours of labor or amount of emissions saved by each alternative are also calculated. These savings are calculated with respect to the reference scenario previously defined. Finally, an analysis comparing the differences in costs and emissions for each alternative is made.

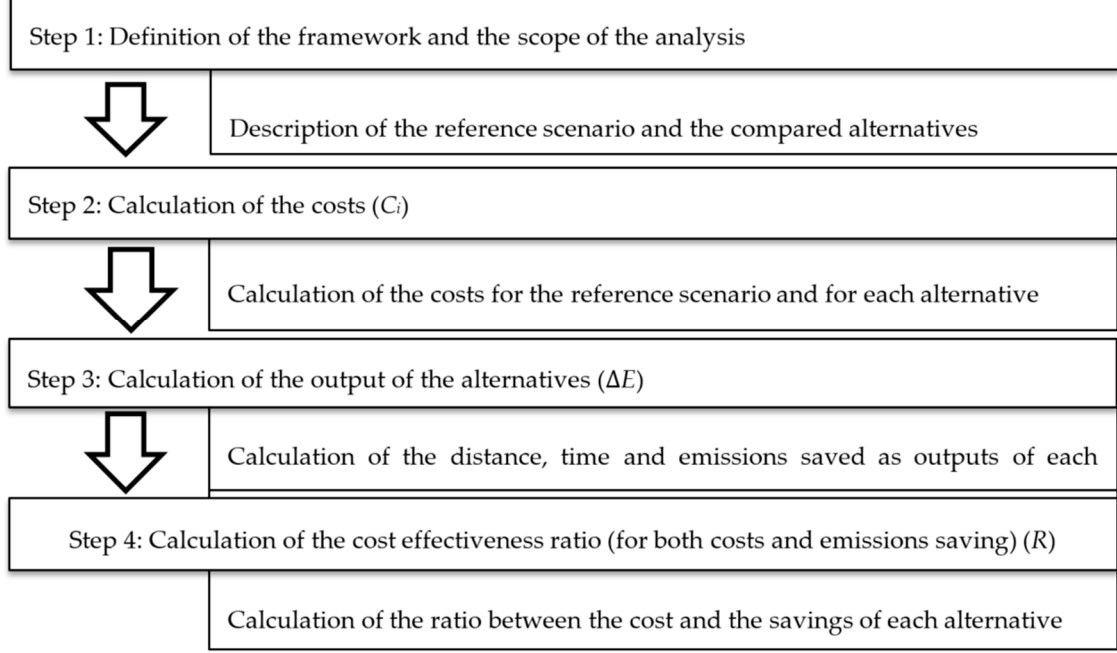

**Figure 6.** The steps undertaken for applying the cost effectiveness methodology. Source: own composition based on [26].

### 2.2.3. CEA Output

CEA is adapted for the analysis of actions in which expected outcomes are clearly identified. If the outcome of a project cannot be clearly defined, or if homogeneous and quantifiable units cannot be determined, the use of cost-effectiveness analysis should be avoided [26]. For example, when an investment aims at reducing the amount of air pollutants that are released in the atmosphere, the effectiveness criteria for that investment could be the decrease in the daily average amount of air pollutants emitted. Other criteria may be more relevant depending on the context of the project.

As such, the CEA output is a ratio between the costs and the outcomes of a project. One has to consider the actualization of investments and depreciation. A more elaborated way of using the CEA is to calculate the ratio between the incremental costs $\Delta C$ and the incremental outputs $\Delta E$ of a project, as shown in Equation (1). This ratio gives information about the cost difference, which is paid to receive the extra, beneficial, output $\Delta E$ (EC, 2008). The incremental cost in some cases can be substituted with the total expenses (costs) $C_f$ of each project. This situation can occur when the outputs of different variants are "compared" with a zero scenario (when $C_i$ is equal to zero): in this case, the resulting ratio is interpreted as the total cost paid to receive the full benefit $\Delta E$ [12].

$$R = \frac{\Delta C}{\Delta E} = \frac{C_f - C_i}{E_f - E_i} \tag{1}$$

### 2.3. CEA Computation and data

The road transport sector is a very competitive market with low profit margins. Due to this, values regarding the cost and benefit of private operators have a confidential status. For this reason, this section presents in detail the computation steps needed to conduct CEA and the publicly available information with regard to this case.

### 2.3.1. Cost Calculation of Road Transport

Blauwens et al. [22] define time and distance costs as the main elements contributing to the expenditures of a transport firm to perform their activity. They present the total cost of a transport task

as being equal to: *uU* + *dD* + *k* + *h*. The notations used are as follows: u—time coefficient; U—total time a vehicle needs for a task; d—distance coefficient; D—total distance covering outwards and return journey; k—tolls, equipment use fees, port dues, etc; h—costs elements which are both depending on time and distance (equipment depreciation and maintenance costs). Each cost element and coefficient is further detailed and examples are given.

Gérard et al. [44] give a complete overview of the costs calculation for road freight transport. The coefficients determined by Blauwens et al. [22] have an average composition and do not completely serve the aim of this study. Because this study focuses on the activity of one transport firm, by making use of these normalized coefficients, the differences in costs created by each type of chassis would be flattened out. To avoid this limitation, the main categories of costs are kept as defined by Blauwens et al. [22], but the cost coefficients are refined taking into account the specific characteristics of the case study and the outcome of interviews.

Finally, the cost for one individual transport journey is calculated with the following formula:

$$C_i = u * U_i + d_l * D_l^i + k_i + h_{ci} \tag{2}$$

where:

$C_i$—cost of journey $i$;

$u$—time coefficient;

$U_i$—amount of hours necessary to complete the transport journey $i$;

$d_l$—distance coefficient calculated according to truck's load as in Table 3, where $l$ is included in (1, 4) $d_l = g_l * p/100$;

$D_l^i$—distance according to vehicle load class $l$ for journey $i$;

$k_i$—amount payable in road tolls for journey $i$;

$h_{ci}$—truck and chassis usage costs for each category of chassis $c$ necessary to complete the journey $i$.

The theoretical framework developed by Blauwens et al. [22] puts forward an in-depth methodology with regard to the time and distance coefficients calculation in determining the operational cost of a transport company. Their model shows that the hour coefficient varies according to the wages of crew, the annual insurance premiums for vehicles, the rent for working spaces and even the general administrative costs of running a fleet of vehicles. Similarly, the distance coefficient depends on the fuel consumption, maintenance, eventual fines and damage liabilities. For the purpose of this research only, the allocation of costs elements is done for the time and distance coefficients. This approach is validated through interviews with representative of the company providing the data for this research and differs from the method proposed by Blauwens et al. [22] as the costs being recognized nor time nor distance-related are included in an extra cost element $h$, referred to as truck and chassis usage cost. This approach considers the time coefficient u as dependent only on the staff cost, while the driver cost is calculated as the average hourly driver salary within the studied company.

The distance coefficient $d$ is directly proportional with the fuel consumption of the vehicle ($g_l$) and the fuel price ($p$). To compare the effectiveness of different chassis, here, several distance coefficients need to be defined according to the carried loads. The practice of reducing the driven distance by consolidating several transport tasks in a single round-journey implies carrying different loads during the round-trip. As such, the loading factor has a direct influence on the total fuel consumption of the vehicle. Table 3 presents four distance coefficients calculated according to vehicle's loading factors as validated by the firm's representative.

**Table 3.** Distance coefficient used in the further analysis. Source: own compilation based on interviews.

| Distance Coefficient | | Load Carried [tons] | Average Fuel Consumption $gl$ [l/100km] | Price per Liter of Fuel $p$ [Euro] | Distance Coefficient [Euro/km] |
|---|---|---|---|---|---|
| $d_1$ | Empty chassis | 0-5 | 28 | | 0.33 |
| $d_2$ | Half loaded container | 5-12 | 30 | | 0.36 |
| $d_3$ | Fully loaded chassis 1 container | 12-32 | 32 | 1.2 | 0.38 |
| $d_4$ | Fully loaded chassis 2 containers | >32 | 34 | | 0.40 |

Costs coefficient $k$ incorporates costs that depend on the route selection such as tolls.

The truck and chassis usage cost $h$, considers depreciation, fines, insurance premiums, maintenance and general administration costs According to the example presented by Blauwens et al. [22], taking into account a sufficiently long period (five to six years for road transport equipment), these elements can be estimated. For this reason and only for the purpose of this study, comparing the use of different chassis, these costs are determined from the daily estimated operational costs put forward by the company for each type of chassis. Table 4 gives an overview of these costs.

**Table 4.** Usage and maintenance costs for each type of chassis used. Source: own compilation based on interviews.

| Cost of Chassis Usage | Platform Chassis | Multipurpose | Mini Eco-combi | Tilt |
|---|---|---|---|---|
| $h_c$ **[Euro/day]** | 35 | 50 | 65 | 45 |

### 2.3.2. Decision Logic Scheme for Using Alternative Chassis

The fleet management practice to reduce the operating costs used by the trucking company is based on a mixed decision process. This process is partially based on the transport planner's experience and a structured decision scheme. The goal is to combine individual transport journeys in one round-trip in order to reduce unproductive movements, as shown in Figure 7 (empty trips).

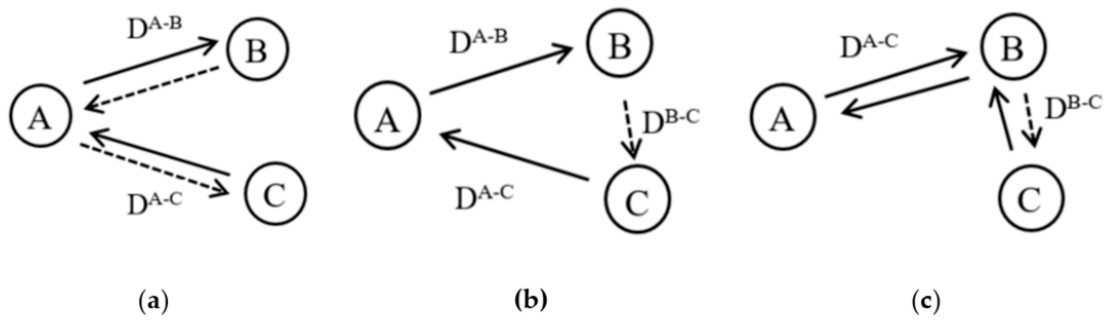

**Figure 7.** Theoretical method for combining two transport journeys in one trip. (**a**) Individual transport journeys per tasks, (**b**) Round journey using multi-purpose chassis, (**c**) Round journey using mini eco-combi chassis.

The first step is to seek for transport tasks that have a similar or neighboring origin and destination. In other words, the distance "$D^{B-C}$" (see Figure 7) is a predefined maximum distance range between two origins/destinations set by a planner to look up for follow-up transport tasks (condition 1). Further steps are set to assign specific chassis to carry out round trips. These steps take into account the characteristics of the two combined tasks (distances between origins and destinations, hours of loading/unloading, quantities to be transported etc.), as detailed in Figure 8. A common example for this case, is the combination of import–export orders (not exclusively), which on a regular basis requires the use of two separate trips to bring, respectively collect the containers. Further in the decision scheme, condition 2 refers to the drop-off time of two consecutive transport tasks *i* and *j*, respectively. This condition determines which type of round trip is going to be chosen and which chassis needs to be used for this combination of tasks. Condition 3 verifies that the time interval between the drop-off and the next pick is sufficient and the latter condition 4 indicates whether the distance between the drop-off location and the pick is close enough so as the combination of transport journeys to be profitable.

The following notations are being made:

- $D^{A-B}$, $D^{A-C}$, $D^{B-C}$—distance between first pick-up and drop-off location, distance between two successive pick-up and drop-off locations and distance between first drop-off point and second pick-up location respectively;
- $t^i_{drop}$, $t^j_{pick}$—time of first drop-off and time of second pick-up respectively;
- $v$—is the average speed;
- $r$—average tariff per kilometer charged;
- $t^j_{op.drop}$—extra operational time needed at drop-off for task *j*;
- $t^j_{op.pick}$—extra operational time needed at pick-up for task *j*.

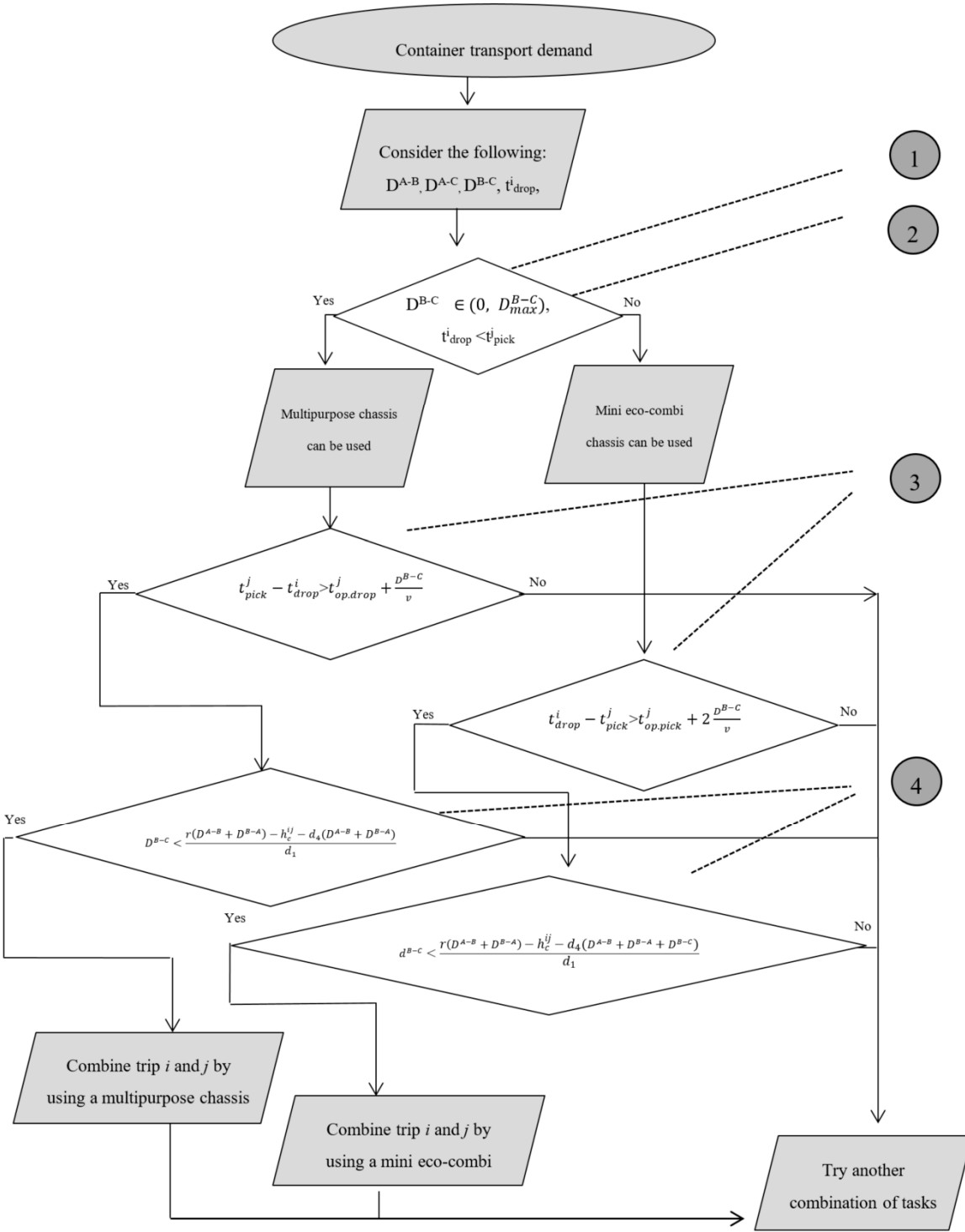

**Figure 8.** Logical scheme representing the decision process for building-up round-journeys.

### 2.3.3. Emission Calculation

Road transport is usually is pointed as the cause of most external emissions. Literature review shows that there are a series of elements that need to be taken into account when calculating the emissions of transport activity. In this matter, a comprehensive review is given by Kok, Annema and van Wee [45]. As well, from a road transport perspective, Piecyk and McKinnon [46] put forward the results of a large scale Delphi survey where they build three scenarios with respect to $CO_2$ emissions levels in freight transport by 2020. Demir, Bektaş, and Laporte [47] make an analysis of models used to

determine the emissions in road transport. Obviously, the total emission is a function of the travelled distance. They point out that, among others, the total vehicle weight, the type of fuel used, and the speed are the most commonly used variables in academic research that address road emissions.

Researchers [48–50] have developed several methods to calculate the emissions coming from transport and point to different types of environmental emissions like carbon dioxide ($CO_2$), tetrahydrocannabinol (THC), non-methane hydrocarbons (NMHC), nitrogen oxides ($NO_x$) or particulate matter (PM). Appendix B gives a non-exhaustive overview of reports and the methods used to determine the emissions of $CO_2$ for road transport. A first observation is made with regards to the fact that $CO_2$ emissions in road transport are calculated either with respect to the weight transported over distance, according to the distance it travels or as a function of the amount of fuel consumed (thus in function of the equipment used). For each method, an emission factor is used in the computations. As well, a special distinction is made between whether the method is applied either to a web tool [51,52] or by research studies [53–55]. In the former case, the input method allows for more flexibility and the outcome is given in relation with the weight transported. Secondly, it is clear that the emission factor calculated only as a function of distance has evolved over time. Regulation regarding emissions in road transport has become more severe. In parallel, manufacturers from the automotive industry have made important steps in developing more fuel-efficient engines that are also less polluting. The overview in the Appendix B with regards to emission factors used in road transport studies shows that researchers have followed the milestones introduced by Euro norms [56].

Conditioned by data availability, this research proposes to quantify the emissions of road transport for each transport journey taking into account the emission coefficient expressed in grams of $CO_2$/veh*km. Facing constraints with regard to data availability, research studies such as Protocol [57] or Cefic [53] present the results regarding emissions as averages of distance for each vehicle. This way, the total amount of $CO_2$ emissions calculated for a transport journey is determined with the following formula [53,57].

$$E_{i-j} = e * D^{i-j} \tag{3}$$

where:

$E_{i-j}$—total amount of $CO_2$ emissions for transport journey $i - j$;

$e$—$CO_2$ emissions coefficient expressed in g $CO_2$/tonne*km and its recommended value to be used in road transport operations is 62 g $CO_2$/tonne*km based on an average load factor of 80% of the maximum vehicle payload and 25% of empty running [53]. For the purpose of this study it is chosen in accordance with [53,54], so the following values are used: 212 g $CO_2$/vehicle*km for vehicles carrying loads between 0-5 tons, 646 g $CO_2$/vehicle*km for vehicles carrying loads between 5-12 tons, 1056 g $CO_2$/vehicle*km for vehicles carrying loads between 12-32 tons and 1254 g $CO_2$/vehicle*km for vehicles carrying loads up to 44 tons.

$D^{i-j}$—distance accordingly to journey $i-j$.

### 2.3.4. Savings Gained Using Innovative Road Solutions (Operational Costs and Emissions)

The calculations with regards to savings gained by using road innovative solutions are done as part of the CEA. The scope of analysis is to determine whether the decrease in operational costs leads also to environmental emissions drops. These calculations are done taking into consideration the distance travelled in each transport journey, the amount of time necessary, the road tolls, the loads that need to be transported and the cost of using each type of chassis. Table 5 details how the costs and emissions are calculated for each alternative. Each calculation practice is derived from Equation (1).

The transport cost of each transport journey, including the reference scenario, is calculated independently. Taking into account the type of chassis, which is being used, specific fuel consumption averages are differentiated among for loaded or empty transports. Other costs, like tolls or chassis usage are also taken into consideration in the cost formula.

After determining the total cost for carrying out each round-trip, the analysis focuses on determining the savings. The savings are also calculated in comparison with the reference scenario.

Table 5 shows, for each round-trip, how the cost and emission savings are calculated. The notation used in the formulas should be read as follows:

$D^{i-j}$—distance from *i* to *j*;
$d_l$—distance cost coefficient according to carried load;
$u$—hour cost coefficient;
$k^{i-j}$—toll fees for route *i* to *j*;
$h_{ci}$—cost of using the chassis *c* for route *i*;
$u^i$—loading/unloading time (for multipurpose chassis) at point *i*;
$u^i_{eco}$—loading/unloading time (for mini eco-combi chassis) at point *i*;
$e$—emission coefficient in g $CO_2$/vehicle*kilometer according to carried load;
$v$—average speed;

*i* and *j* keep place for locations' origins and destinations, and are indicated as locations A, B, or C through the reference scenario.

**Table 5.** Comparison of chassis usage: calculating the costs and emissions savings. Source: own composition.

| Transport Practice | Type of Round-trip | Total Savings | | Emissions Saved (Relative to Reference Scenario) |
|---|---|---|---|---|
| Reference scenario—the use of two individual trucks for each transport task. | PD, PD | - | - | - |
| Case A. Mini eco-combi chassis—the combination of two containers transported using a mini eco-combi chassis. | PPaDPD | $u*(2*\frac{D^{C-A}-D^{B-C}}{v} + (u_{eco}^l + u_{eco}^u)$ | Time | $e*(D^{B-A} + D^{A-C}$ $+D^{C-A} - D^{B-C}$ $-D^{C-B} - D^{B-A})$ |
| | | $d_l^{B-A}*D^{B-A} + d_l^{A-C}*D^{A-C} + d_l^{C-A}*D^{C-A}$ $- d_l^{B-C}*D^{B-C} - d_l^{C-B}*D^{C-B}$ $- d_l^{B-A}*D^{B-A}$ | Fuel consumption | |
| | | $k^{B-A} + k^{A-C} + k^{C-A} - k^{B-C} - k^{C-B} - k^{B-A}$ | Tolls | |
| | | $+2h_{ci} - h_{cj}$ | Chassis usage | |
| Case B. Multipurpose chassis—the combination of two containers transported performed using a multipurpose chassis (ex: new steer-chassis). Case C. Tilt chassis container—the same container is used to respond to the second transport task. Case D. Re-use of empty container—the same container is used to respond to the second transport task. | PDPD (PULD), PPDD (1), PPDD (2), PPDD (3) | $u*\frac{(D^{B-A}+D^{C-A}-D^{B-C})}{v}$ | Time | $e*(D^{B-A} + D^{C-A} + D^{B-C})$ |
| | | $d_l^{B-A}*D^{B-A} + d_l^{C-A}*D^{C-A} + d_l^{B-C}*D^{B-C}$ | Fuel consumption | |
| | | $k^{B-A} + k^{C-A} + k^{B-C}$ | Tolls | |
| | | $2h_{ci} - h_{cj}$ | Chassis usage | |

## 3. Results

The CEA applied in this paper is done based on a real set of data coming from a road transport company. The used data set contains information over the transport tasks that have been combined in round–round journeys. This data refers to transport deliveries of 20′ containers. Detailed information is used regarding the loaded/unloaded status, location, time of pick-up/drop-off and the type of chassis used for each journey. One must be aware that the hard data with regard to each individual trip is the object of a bilateral agreement between the data provider and the researchers conducting this study, being thus not publicly available.

Data covering two periods was used in two analyses. The first analysis focuses on a long period of five months from 1 March to 31 July 2015. As well, this period is used as main sample and it ensures that the combination of transport tasks is consistent. These transport journeys are used to derive the main conclusions of this paper. In addition, the second analysis focuses on a shorter time interval of two weeks, from 14 to 25 April 2014. This period overlaps with data for the same time interval from the initial sample. The latter will show whether the use of the same practice has changed from one year to another and whether there is impact by seasonality of transport practices.

In this study, the data collected shows that in the period of five months, a total of 2992 round transport journeys with an average length of 462 km were performed. Figure 9 gives an overview of the used transport practices for the road journeys. The highest proportion of tasks has been done by using mini eco-combi chassis (56%), followed by multipurpose chassis (23%), re-use empty containers (19%) and tilt chassis (2%).

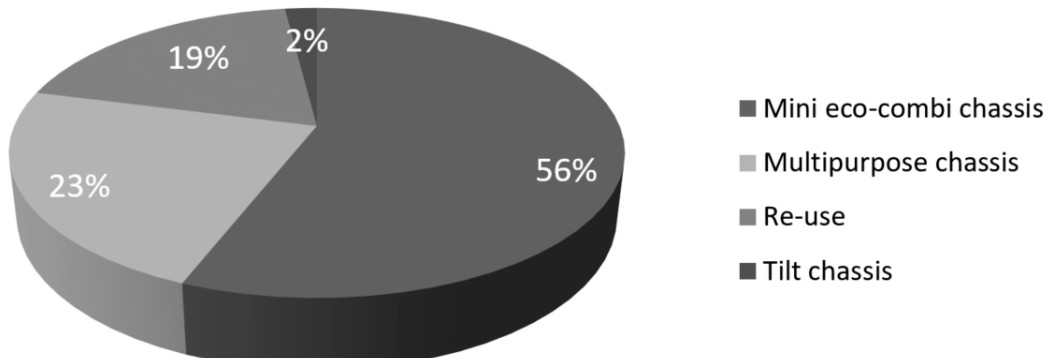

**Figure 9.** Share of practices used by the road transport company in its operations. Source: own calculation.

A main remark is that the mini eco-combi chassis is the most popular option for adding two or more transport tasks in one journey. The re-use of empty container and the multipurpose chassis have an approximately equal share in the daily operation of the studied data set.

Figure 10 presents a detailed overview of the operational cost and $CO_2$ emissions for each transport practice. The main elements that take part in the cost structure of road transport are labor and fuel consumption costs. These costs and $CO_2$ emissions are determined as a function of each journey length. As it can be seen in Figure 10, these two outcomes vary similarly for each type of transport practice.

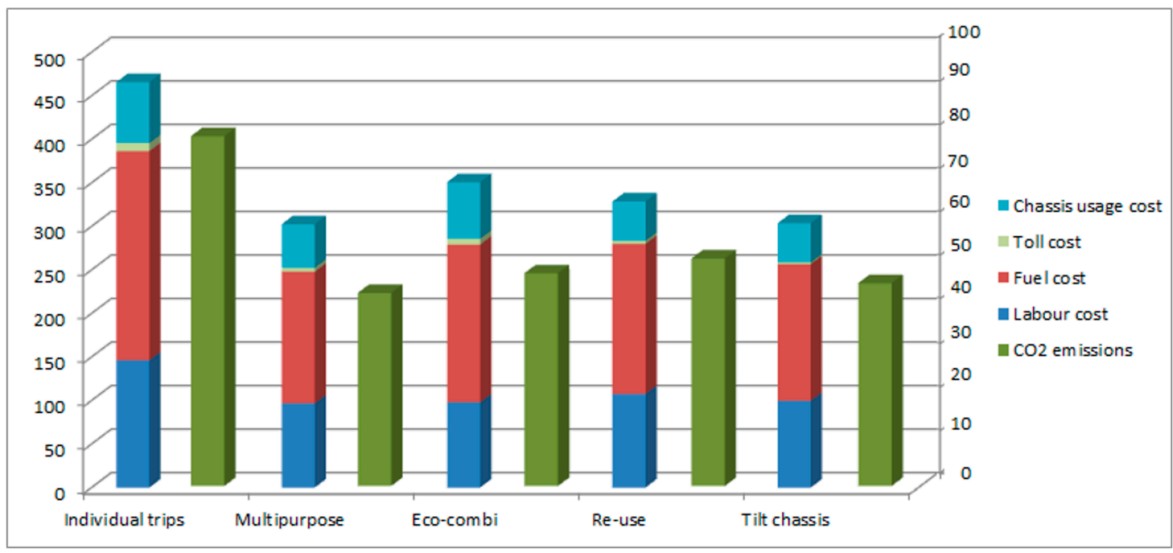

**Figure 10.** Total cost and emission of each transport practice. Source: own calculations.

A more detailed overview of the savings generated by using innovative solutions in road transport operations is presented in Table 6. The percentages showing the costs and emission savings are calculated relative to the reference scenario.

**Table 6.** Costs and emissions saving for each trip category. Source: own calculations.

|  | Individual Trips Scenario | Round-journey Practice | Percent Reduction |
|---|---|---|---|
| **Multipurpose** |  |  |  |
| cost [Euro] | 444.9 | 289.7 | 35% |
| emission [kg/trip] | 78.60 | 48.95 | 38% |
| **Mini eco-combi** |  |  |  |
| cost [Euro] | 468.03 | 350.77 | 25% |
| emission [kg/trip] | 80.77 | 53.05 | 34% |
| **Re-use** |  |  |  |
| cost [Euro] | 482.45 | 322.64 | 33% |
| emission [kg/trip] | 85.97 | 56.07 | 35% |
| **Tilt chassis** |  |  |  |
| cost [Euro] | 471.45 | 310.98 | 34% |
| emission [kg/trip] | 84.01 | 54.05 | 36% |

From Table 6, two conclusions can be derived. With regards to the savings generated by using each transport practice, firstly, combining transport tasks in one journey reduces the operational cost by between 25% and 35%. Secondly, the $CO_2$ emissions are reduced as well. On average, the $CO_2$ emissions are lower by 34% to 38% in the case of round-journeys. The highest costs and $CO_2$ emission reduction is achieved by using chassis from the multipurpose category. The computed values there are on average by 35% lower in case of costs and by 38% lower in case of $CO_2$ emissions. In contrast, the outcomes with respect to the eco-combi show that the operational costs are lower by 25% for costs and by 34% for the $CO_2$ emissions.

These conclusions are further confirmed also by the CEA outcomes. Table 7 presents for each category of round-trips the costs and the amount of $CO_2$ saved. A further ratio between the cost of a round-trip and the $CO_2$ emissions saved determines the cost effectiveness ratio with respect to each transport practice. Using the multipurpose chassis is the most cost-effective practice, followed by the re-use of empty containers and eco-combi chassis.

**Table 7.** Environmental effectiveness of each type of transport practice. Source: own calculations.

|  | **Multipurpose** | **Mini Eco-Combi** | **Re-Use** | **Tilt Chassis** |
|---|---|---|---|---|
| Average cost per round-trip [Euro/round-trip] | 289.7 | 350.8 | 322.0 | 310.9 |
| Emissions saved [kg/round-trip] | 29.6 | 27.7 | 29.9 | 29.9 |
| Cost effectiveness [Euro/kg emissions saved] | 9.8 | 12.7 | 10.8 | 10.4 |

Triggered by these results, a comparative analysis regarding the data form the same period of different years is conducted. A sample of round-journeys done in the period 14-25 April 2014 is compared with the same period (13–24 April) in 2015. Figure 11 shows the types of transport practices that have been used in the same period of two years. It is noticeable that no significant changes have occurred over years. Moreover, for each category of round-trips, the cost and $CO_2$ savings are presented in Table 8. These elements are further analyzed in relation to the length of the round-journeys.

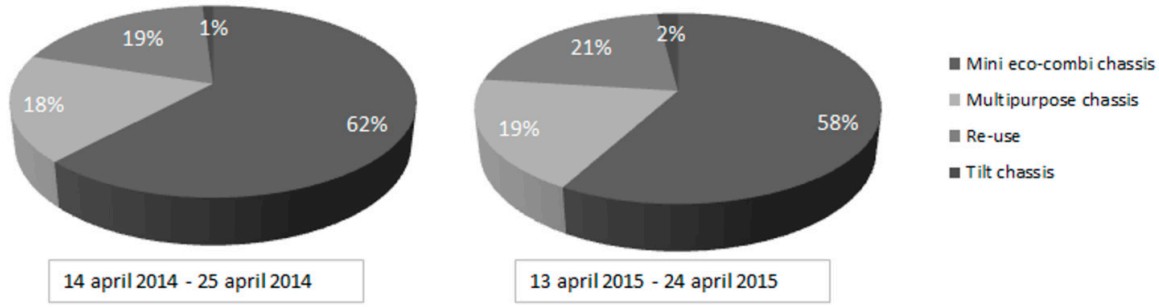

**Figure 11.** Evolution of share of transport practices used over years. Source: own calculations.

**Table 8.** Comparison of each type of transport practice usage over two time intervals. Source: own calculations.

|  | Cost [Euro/Journey] | Cost Saving [Euro/Journey] | Emission Saving [kg/Journey] | Total Distance travelled per Trip [km] | Distance between Intermediary Points [km] |
|---|---|---|---|---|---|
| **Multipurpose chassis** |  |  |  |  |  |
| 14th–25th April 2014 | 321.7 | 175.6 | 54.4 | 432 | 82 |
| 13th–24th April 2015 | 318.7 | 181.0 | 56.6 | 423 | 75.6 |
| Variation | −1% | 3% | 4% | −2% | −8% |
| **Eco-combi** |  |  |  |  |  |
| 14th–25th April 2014 | 375.3 | 144.9 | 57.0 | 398 | 58.3 |
| 13th–24th April 2015 | 302.8 | 114.5 | 42.8 | 348 | 62 |
| Variation | −19% | −21% | −25% | −12% | 6% |
| **Re-use** |  |  |  |  |  |
| 14th–25th April 2014 | 325.4 | 165.0 | 56.2 | 470 | 102 |
| 13th–24th April 2015 | 318.9 | 175.6 | 61.3 | 459 | 96.2 |
| Variation | −2% | 6% | 9% | −2% | −6% |
| **Tilt chassis** |  |  |  |  |  |
| 14th–25th April 2014 | 331.7 | 155.7 | 57.6 | 463 | 97 |
| 13th–24th April 2015 | 326.5 | 159.4 | 64.3 | 455 | 93 |
| Variation | −2% | 2% | 12% | −2% | −4% |

With regards to the cost of each type of chassis, firstly, the average cost of a transport journey has decreased from one year to another regardless the practice used. This is explained by the decrease in the total distance travelled for each trip. Concerning the usage of mini eco-combi chassis, the cost savings did not improve over years. An explanation can be found in the fact that the distance between

intermediary loading and unloading points increases. In other words, for this type of chassis, regardless the previous year, the transport tasks chosen to be added in the same round journey were further located form each other. This stretch, by having a higher share in the total travelled distance, lowered the performance with regards to fuel consumption and labor costs. As these results demonstrate, choosing to combine transport tasks for which the successive destination and origin are not located in the vicinity of each other has a negative effect on the cost effectiveness. This remark is supported by the opposite results in case of multipurpose chassis and re-use of empty containers. For these practices, the transport tasks that have been chosen have closer destination and origin points, and by comparison, have a better cost effectiveness outcome.

## 4. Conclusions

Road transport operators need to innovate forced by a competitive environment and a low profit margins sector. Most innovation developments in the road sector focus on economic benefits. Nonetheless, environmental objectives are necessary nowadays. Innovation in the road transport sector has positive consequences also on the environment. However, this type of impact is incidentally achieved, and thus not always acknowledged. To address this shortcoming, the paper applies a cost-effectiveness analysis on a road transport case study to prove environmental merits. Such technological innovation (e.g., innovative chassis use), even though not subject to incentives from policy makers, has a high potential for environmental benefits. Moreover, these technological achievements have a high impact on the global effects of supply chains and freight transportation systems. After an in-depth literature review of studies dedicated to CEA, an investigation about the characteristics of innovative solutions used in road transport of containers in the hinterland of a port in Western Europe is carried out. These solutions are the object of the CEA, which has the goal to gain more insight into the decision-making process and to evaluate how these decisions ultimately impact environmental quality.

This study adds to the current state of science through its detailed methodological approach. This methodology puts forward the process through which dispatchers take operational decisions in a detailed theoretical model. Moreover, the calculation process of costs, monetary savings, and emission reduction generated by innovation in transport operations is presented in a detailed overview. As opposed to similar studies, the cost-effectiveness ratio shows, in a composite unit, the performance of several transport practices.

Moreover, this research provides evidence on the practice of a firm active in the road transport sector to adopt clean transportation planning practices and technologies if the outcomes are boosted by economic performance and/or governmental financial support. This hypothesis is also tested by the current approach. Starting thus from a real set of data, this paper investigates whether innovative road transport practices that have pure economic objectives bring also environmental benefits.

The case study analysis focuses on practices used for transport of 20′ containers in import/export operations at the hinterland of a seaport where tasks consolidation in round-trips is possible. A further assumption is that any transhipment or container swap-movement is not possible at locations in hinterland as no equipment (e.g., cranes, reach stackers) is available. Therefore, four road transport practices are being analyzed in relation with a reference scenario. These technological innovation initiatives are introduced into a chassis management scheme that has the purpose to combine two or more transport tasks in one round-journey. Involved are the multipurpose chassis, the mini eco-combi, the tilt chassis and the re-use of empty containers. For each type of chassis, the costs and emissions savings brought by the round journey are quantified using particularized functions.

The results show that by using different innovative chassis and new planning procedures to form round journeys, a transport company has positive achievements with respect to both costs and environmental emissions. While the costs are reduced on average by 25% to 35%, the environmental emissions are lowered by 34% to 38%. Moreover, the use of the multipurpose chassis represents the most cost-effective practice that addresses environmental emissions. A further conclusion is made

with regards to the location of successive origins and destinations of transport tasks that are combined in one round journey: increasing the distance between successive origins and destinations makes the combination of transport journeys less effective.

This research is relevant for industry as well for researchers conducting studies in road transport. The methodology presented, results and interpretation represent a basic foundation on which operational decisions in road transport can be made. Moreover, this research shows to policy makers that technological innovation in road transport brings environmental benefits as well. The latter could benefit from higher appreciation. This research's results are given from the perspective of one road transport operator, for practices applicable for transport of 20′ containers and for operations in the hinterland area of a seaport that involves import/export container movements. The methodology used can be generalized globally for this global niche of road transport operations.

Therefore, further research is required to validate the findings from this paper more in depth and for other cases. The expansion of the dataset to a longer period would allow for consistency testing. Similar research can be carried with extending the round-trip possibilities enabled by including 40′ containers. Eco-combi chassis that allow the transport of three 20′ containers or one 40′ and one 20′ container could be transported in the same journey. Equally, the opportunity cost in relation to the human resources when calculating the cost of different transport options could be included. It would be useful to see as well whether the findings of this paper are confirmed also by a larger set of transport journeys or other innovative transport practices.

**Author Contributions:** Conceptualization, V.C., C.S., T.V.; Methodology, V.C.; Data curation, V.C.; Formal analysis, V.C.; Writing—original draft preparation, V.C.; Writing—review and editing, V.C., C.S. and T.V.

**Funding:** This research was funded by the BNP Paribas Chair of Transport, Logistics and Ports at the University of Antwerp, under the grant/contract number FFA4872.

**Acknowledgments:** The authors would like to thank for sharing his professional experience to Kurt Joosen, CEO Transport Joosen, and Yves Haudhuyze, COO Transport Joosen.

**Conflicts of Interest:** The authors declare no conflict of interest and that the funding entity had no role in the design of the study; in the collection, analyses, or interpretation of data; in the writing of the manuscript, or in the decision to publish the results.

## Appendix A

**Table A1.** Supply chain and ports innovation typology.

| Innovation Typology | Description |
| --- | --- |
| I. Technology—unit change | A primarily technological change occurring at one specific location and/or for one specific operator |
| II. Technology—market change | Like I, but the change occurs for an entire product market (e.g., container handling) |
| III. Technological, Managerial, Organizational, Cultural—business change | Next to technological changes, the innovation also allows for changes at managerial, organizational, and cultural level, all of those at the level of a specific business |
| IV. Technological, Managerial, Organizational, Cultural—market change | Like III, but the change occurs for an entire product market |
| V. Managerial, Organizational, Cultural—market change | Innovation into the organizational culture and management processes without significant technological component |
| VI. Policy initiatives (Managerial, Organisation, Cultural—market change) | Policy-initiated innovation actions, which in turn may trigger further innovation. (e.g., introducing carbon tax) |

Source: Vanelslander et al. [21].

## Appendix B

**Table A2.** Emission coefficient expressed in grams $CO_2$ for heavy trucks (Euro VI) used either in web tool applications or research studies.

| Grams $CO_2$/tonne*km | Grams $CO_2$/vh*km | Grams $CO_2$/Litter of Fuel | Date Application (Geographic Area) | Obs. | Source |
|---|---|---|---|---|---|
| Web tools | | | | | |
| 51,2 | | | | | [52] |
| 65 | | | | | [51] |
| 81.48 | | | | | [58] |
| 84 | | | | | [59] |
| 86 | | | | | [60] |
| Research studies and reports | | | | | |
| 62 | | 2900 | March 2011 | | [53] |
| 82 | | | 10 June 2015 | Value for tractor and trailer/chassis | [61] |
| 129 | | | 2009 | | [55] |
| 100 | | | 2009 | | [62] |
| | 870 | | 2005 (US) | Light commercial vehicle | [57] |
| | 588–818 | | 2009 (Europe) | Light commercial vehicle | [63] |
| | 190 | | 2011 | Belgium average all types of roads for personal vehicles | [64] |
| | 120 | | 2014 (EU) | Personal vehicles | [54] |

Source: own compilation.

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
