# Peer review of "Innovation in Road Freight Transport: Quantifying the Environmental Performance of Operational Cost-Reducing Practices"

_sustainability, doi:10.3390/su11082212_

Round 1
Reviewer 1 Report
There are several errors or unclear parts of the analysis that raise doubts:
1. The object of the analysis are existing and well-known construction solutions of container semitrailers. Relatively new products are D-TEC Flexitrailer and load-shifting chassis. Still it can be treated as INNOVATION in road transport. Especially, ‘"re-use’ of empty containers" is NOT a technical innovation.
2. The analysis is narrowed to 20' con
2. The scenario called "leg-mounted chassis" is unclear. On the market apart of containers swap bodies are used. It is STANDARD intermodal unit and transport technology. The only need is the pneumatic suspension of the semitrailer. Is it the case?
3. On table 2:D) (P) Pick-up should be distinguished from loading and (D) Drop should be distinguished from unloading. It is great difference. One notice: on standard semitrailer unloading of he front 20' containers is impossible without moving the rear 20'.
4. It should be clearly stated that the analysis takes under consideration only 20' or short swap bodies without side doors (courtines) and any transhipment by cranes or reachstackers are unavailable at hinterland clients.
5. There are unexplained indications on Figure 7.
6. Formula (1) is equalling incremental CEA ratio and non-incremental one. It is not justified.
7. On table 6 "Individual trips scenario" is based on standard semitrailers or not? Why it differs between cases?
8. On table 7 are not "percentages in saving: are presented?!
9. The conclusions are too wide in relation to the scope of research (see point 4).

Author Response
1. The object of the analysis are existing and well-known construction solutions of container semitrailers. Relatively new products are D-TEC Flexitrailer and load-shifting chassis. Still it can be treated as INNOVATION in road transport. Especially, ‘"re-use’ of empty containers" is NOT a technical innovation.
Response: Text has been corrected taking into account this comment.
The focus of the paper is set on both technical innovation but also on the planning process. These changes are introduced through a new managerial and cultural (change of mind set) driven innovation who from his user position thinks together with the supplier. This practice is new and innovative in the road transport sector. The current content was adapted to clearly reflect this specificity. The re-use of empty containers fall under this type of business driven organisational innovation. Although these solutions are in use and known to the stakeholders active in this sector, this innovation in road transport does not cross this sector’s borders and is unknown to the wide public. For this reason the road transport stakeholders get the label of un-innovative sector. These solutions are worth to be treated as innovation indeed and be considered as case studies for this research.
2. The analysis is narrowed to 20' con
Response: Indeed, this element is now clearly specified in the paper.
2. The scenario called "leg-mounted chassis" is unclear. On the market apart of containers swap bodies are used. It is STANDARD intermodal unit and transport technology. The only need is the pneumatic suspension of the semitrailer. Is it the case?
Response: Yes it is the case where drivers make use of both swap bodies capabilities and pneumatic suspension for repositioning the container on the trailer. The text “the leg-mounted chassis operations need extra time to fix the container on the mounted legs and reposition it on the trailer” was reformulated to “the leg-mounted chassis need extra time to expand/retract the mounted legs, to set in use the pneumatic suspension of the semitrailer and reposition it on the trailer.”
3. On table 2:D) (P) Pick-up should be distinguished from loading and (D) Drop should be distinguished from unloading. It is great difference. One notice: on standard semitrailer unloading of he front 20' containers is impossible without moving the rear 20'.
Response: Thank you for the suggestion, the “loading of a container” in text refers to the action of loading a container on a chassis and not physically loading the good inside it. This differences are factually included in the calculations, thought the coefficients used for each type of operation. Further explanations are included now in the paper. E.g. for both P and D the following adding was done: “pick-up or loading of a container on a chassis and drop-off” or “unloading of a container from a chassis”, new indications for loading and unloading operations were added.
4. It should be clearly stated that the analysis takes under consideration only 20' or short swap bodies without side doors (courtines) and any transhipment by cranes or reachstackers are unavailable at hinterland clients.
Response: Thank you for your suggestion. A further explanation was added in section 2.1.2 and in table 2 to clarify this matter.
5. There are unexplained indications on Figure 7.
Response: The paper was restructured as such that section 2.1.4. (containing figure 7) was moved to section 2.3. and included in the paper as section 2.3.2. This way the figure 7 and the related indications follow section 2.3.1. that explains the elements considered in figure 7.
6. Formula (1) is equalling incremental CEA ratio and non-incremental one. It is not justified.
Response: A correction was added so that the incremental CEA ratio equals an incremental one. For the non-incremental one, explanations were added in the text.
7. On table 6 "Individual trips scenario" is based on standard semitrailers or not? Why it differs between cases?
Response: The calculations for each case are made based on trips carried using each type of chassis. The results with regard to the “individual trips scenario” are calculated from the perspective that the trips would be carried according to this reference scenario.
8. On table 7 are not "percentages in saving: are presented?!
Response: The text is revised taken this comment into account.
9. The conclusions are too wide in relation to the scope of research (see point 4).
Response: Clear explanations have been added with regard to the narrow focus of the research. A reference with regard to the limitation addressed in point 4 was added as well.
Reviewer 2 Report
The paper presents a study that aims to quantify the benefits (cost savings, emissions reduction, etc.) from using innovative chassis configurations as part of freight operations. The main limitation of the current version of the paper is the lack of clear contributions to knowledge. This is further elaborated as follows:
The answering of the two Research Questions (ROs) at the bottom of the page 3 is difficult to be justified as a major contribution since the study is based on a single freight operator. Therefore, any generalisation of the results is questionable.
There is a lot of research on freight operations scheduling and cargo management to address the problems stated in the paper (ie, empty cargo kms, efficient journey planning, etc.). The planning stage for optimised scheduling and the potential savings due to that has not been fully appreciated. If the study is only relevant for specific freight operations (ie. import/export -- port/hinterland) then this limits the significance of the results.
If the main contribution of the paper is the methodological approach followed (something alluded in the last paragraph of the paper), then this is not evident in the current version of the paper. Therefore, further work is necessary to explain how this proposed approach excels current practices.
Author Response
The paper presents a study that aims to quantify the benefits (cost savings, emissions reduction, etc.) from using innovative chassis configurations as part of freight operations. The main limitation of the current version of the paper is the lack of clear contributions to knowledge. This is further elaborated as follows:
The answering of the two Research Questions (ROs) at the bottom of the page 3 is difficult to be justified as a major contribution since the study is based on a single freight operator. Therefore, any generalisation of the results is questionable.
Response: Extra explanations have been introduced in the conclusion section to pinpoint the differentiating elements of this research.
There is a lot of research on freight operations scheduling and cargo management to address the problems stated in the paper (ie, empty cargo kms, efficient journey planning, etc.). The planning stage for optimised scheduling and the potential savings due to that has not been fully appreciated. If the study is only relevant for specific freight operations (ie. import/export -- port/hinterland) then this limits the significance of the results.
Response: The text has been enriched to give more credit to the planning process of transport within the hinterland area of sea-port. The importance of this kind of traffic is shown through figure 1.
If the main contribution of the paper is the methodological approach followed (something alluded in the last paragraph of the paper), then this is not evident in the current version of the paper. Therefore, further work is necessary to explain how this proposed approach excels current practices.
Response: Extra explanatory paragraphs were added considering this comment. The new adding points that this paper excel through its detailed methodology by putting forward the theoretical decision process followed by dispatchers in their planning operations. Also, the new adding states that elements this paper, in comparison to other, presents detailed calculation models for cost, monetary saving and emission reductions in road transport planning.
Reviewer 3 Report
Interesting article, however does not contemplate innovative aspects. The contribution is in the way cost analysis has been organized. Although some points are not clear, for example, the choice and construction of the models, the calibration process and their validation.
You need to make clear in the introduction at which points the innovation in cargo transportation is occurring. You are working with a part that involves equipment, but innovation would have other aspects.
It is also unclear the reason for studying CO2 reduction alone, since from the environmental point of view, particulate matter is much more significant in freight transport than CO2.
I believe that these points are aligned with a more elaborate and grounded theoretical revision to the point of explaining why the work is being proposed.
Author Response
Interesting article, however does not contemplate innovative aspects. The contribution is in the way cost analysis has been organized. Although some points are not clear, for example, the choice and construction of the models, the calibration process and their validation.
Response: Further explanations have been provided with regard to both innovative aspects and the construction of the theoretical model.
You need to make clear in the introduction at which points the innovation in cargo transportation is occurring. You are working with a part that involves equipment, but innovation would have other aspects.
Response: Text has been corrected taking into account this comment. The focus of the paper is better defined. These changes are introduced through a new managerial and cultural (change of mind set) driven innovation who from his user position thinks together with the supplier. This practice is new and innovative in the road transport sector. Although these solutions are in use and known to the stakeholders active in this sector, this innovation in road transport does not cross this sector’s borders and is unknown to the wide public and it is now highlights through the content of the paper.
It is also unclear the reason for studying CO2 reduction alone, since from the environmental point of view, particulate matter is much more significant in freight transport than CO2.
Response: Tet was revised considering this comment. The paper acknowledges the fact that environmental effect of transport activities can be quantified through multiple types of emissions e.g. CO2, NOx, PM or SOx. However, the ultimate purpose of this research is to calculate the CEA ratio with regard to emissions for each innovative practice. Through the application of the CEA methodology, the CO2 emission is used as indicator for emissions. This indicator is picked from the literature review and is explained in the methodological approach. Changing the type of emission to which the costs are reported (assuming a similar approach for indicating the emission levels is taken) will not fundamentally change the research’s results shown as percentage of emission reduced nor the comparison between different innovative practices provided through the CEA methodology.
I believe that these points are aligned with a more elaborate and grounded theoretical revision to the point of explaining why the work is being proposed.
Response: Thank you very much for your suggestions. The content of the paper has been enriched to reflect the concerns that were raised.
Round 2
Reviewer 1 Report
I appreciate your responses and amendments. I see the essential improve.
Reviewer 2 Report
The authors have addressed the comments of the previous review. The paper has been substantially improved and can be considered for publication.
Reviewer 3 Report
The presented manuscript brings clear and effective discussion on the analysis of the operational cost from the reduction of environmental impacts caused by the inclusion of different technologies in the operation between ports.
From this second round we notice that aspects initially suggested were incorporated and made the reading clearer.
As a suggestion, since it is not part of the scope of the article, to carry out measurements on board, in order to quantify the exact values of the impacts, since a lot of premises and parameters were used that, not necessarily, configure the situation under analysis.
I no longer believe that the paper is already in a position to bring contributions to the community.